# Comparison of Single-Shot and Two-Shot Deep Neural Network Models for Whitefly Detection in IoT Web Application

**Chinmay U. Parab** [1], **Canicius Mwitta** [2], **Miller Hayes** [2], **Jason M. Schmidt** [2], **David Riley** [2], **Kadeghe Fue** [3], **Suchendra Bhandarkar** [1] and **Glen C. Rains** [2,*]

1   Department of Computer Science, University of Georgia, Athens, GA 30602, USA;
    chinmayumeshparab@uga.edu (C.U.P.); suchi@uga.edu (S.B.)
2   Department of Entomology, University of Georgia, Tifton, GA 31793, USA; cmwitta@uga.edu (C.M.);
    miller.hayes@uga.edu (M.H.); jschmid2@uga.edu (J.M.S.); dgr@uga.edu (D.R.)
3   Department of Agricultural Engineering (DAE), Sokoine University of Agriculture,
    Morogoro 30007, Tanzania; kadefue@sua.ac.tz
*   Correspondence: grains@uga.edu

**Abstract:** In this study, we have compared YOLOv4, a single-shot detector to Faster-RCNN, a two-shot detector to detect and classify whiteflies on yellow-sticky tape (YST). An IoT remote whitefly monitoring station was developed and placed in a whitefly rearing room. Images of whiteflies attracted to the trap were recorded 2× per day. A total of 120 whitefly images were labeled using labeling software and split into a training and testing dataset, and 18 additional yellow-stick tape images were labeled with false positives to increase the model accuracy from remote whitefly monitors in the field that created false positives due to water beads and reflective light on the tape after rain. The two-shot detection model has two stages: region proposal and then classification of those regions and refinement of the location prediction. Single-shot detection skips the region proposal stage and yields final localization and content prediction at once. Because of this difference, YOLOv4 is faster but less accurate than Faster-RCNN. From the results of our study, it is clear that Faster-RCNN (precision—95.08%, F-1 Score—0.96, recall—98.69%) achieved a higher level of performance than YOLOv4 (precision—71.77%, F-1 score—0.83, recall—73.31%), and will be adopted for further development of the monitoring station.

**Keywords:** object detection; convolutional neural network; whiteflies; web app; insect trap; IoT

## 1. Introduction

The infestation of silverleaf whitefly, *Biotype B Bemisia tabaci Gennadius*, is a globally identified challenge when it comes to vegetable and cotton production. Of the crops affected by whitefly, cucurbits are some of the most severely affected and support the highest levels of whitefly reproduction. One of the issues faced by growers is tracking whitefly movement in production areas, particularly at adjacent cotton and cucurbit crops. The timing of whitefly management practices is mostly preventative. Whiteflies can move into a vegetable or cotton field in two major ways. First, there can be a small, overwintering population that build ups over 2–3 whitefly generations before it can reach damaging levels in that field. Secondly, whiteflies can migrate in large numbers from one host crop to another and quickly overwhelm a crop in a single generation. Early warning of such infestation and migration could provide farmers and consultants the appropriate information to apply management strategies in a timely and efficacious manner. Once whiteflies become entrenched in a crop, it is difficult to manage economically significant losses.

Without automated data analysis techniques, detecting and reporting whitefly movement is very laborious and time-consuming. Yellow-sticky cards (YSC) are placed adjacent to fields and/or across broad landscapes. A person must travel to each location and retrieve traps to be brought back to the lab and whiteflies counted manually. This requires many man hours of time in travel and counting. This can also reduce the ability to manage these pests in an environmentally and economically viable way. Whiteflies are best controlled while moving from host to host or across landscapes. Once they land and move underneath the leaves of their host, they are much more difficult to control. Consequently, a trap must be able to make measurements in a timely manner ($1\times$ to $2\times$ a day), be deployed for at least 3 months without maintenance and report this information to farmers, researchers, consultants and extension personnel. This requires the ability to disregard dust and reflections from water droplets when detecting and counting insects. Finally, insect species such as whiteflies, aphids and thrips are extremely small pests and require a system that can continually measure and report infestation or population dynamics of insect species.

Previous whitefly detection studies have been mainly based on image processing to detect objects in an image and then use machine learning algorithms to classify them. A method for counting whiteflies on the leaves of crops was proposed by [1]. In the study, a hypothesis framework was defined to distinguish pests from other particles in an image; then, color, texture and shape analysis was performed. In color analysis, the RGB (Red Green Blue) picture is converted into a gray-scale image. In texture and shape analysis, entropy properties and morphological openings with structuring elements are used. An image processing approach using MATLAB for detecting and counting the number of whiteflies on the leaves of crops was developed by [2]. They first converted the RGB picture to an HSV image and then performed background subtraction and morphological operations to detect whiteflies. Other studies have explored Principal Component Analysis [3] segmentation and the Mahalanobis distance classifier for white fly, aphid and thrips identification [4], machine vision technique for whitefly scouting in greenhouse [5] and discriminant analysis to identify a European vs. Africanized honey bee [6]. Methods have also been developed using artificial neural networks (ANN) with input nodes as features for identifying multiple species of insects such as thrips [7]. They used seventeen continuous salient features and two binary qualitative measurements of eighteen common European species of thrips. These methods are generally semi-autonomous and require some measurements or manual manipulation of data to run the detection algorithm. Consequently, we propose a deep neural network approach that makes it easy to add insect species' detection and identification and increase the utility of the model.

Deep Neural Network (DNN) approaches have been widely used in object detection and identification, especially the Convolutional Neural Network (CNN) algorithm. The CNN algorithm helps extract representations from the image. Unlike non-neural approaches, the CNN algorithm takes raw pixel data, trains the model, and then extracts features automatically for better classification. A convolutional neural network to detect moths and make the model non-species specific so new insect species could be added later was used by [8]. A computer vision algorithm, [9], referred to as Moth Classification and Counting (MCC), based on deep learning analysis of the captured images, tracked and counted the number of insects and identified moth species. Tracking was implemented to reduce the potential for counting moths more than once as they walked around on the light trap. A DNN model cause Squeezenet was developed for the detection of tiger beetles [10]. They found that freezing some of the model layers during training improved accuracy to 90%. An automated monitoring system for fruit flies [11] in crops for pest management using a CNN was developed. It used a trained ResNet model to detect spotted wing Drosophila (Drosophila suzukii). A computer vision detector using Haar Cascade and Deep Learning techniques [12] was developed to automatically count whiteflies on infested cassava leaves.

A phone app [13] was created by AI-Dev Lab Makerere University, Kampala, Uganda, and is available for download on Google Play. It was developed for counting whiteflies on cassava leaves in real-time in the field by taking a picture of whiteflies on plant material. There have been several successful whitefly counting algorithms developed for specified crops. A color segmentation and mathematical morphology to count adults, nymphs, and exoskeletons on soybean leaves was developed by [14]. Others have also employed successful algorithmic approaches to whitefly detection and counting on leaves ([15,16]) that are collected and brought to the lab or images taken with a handheld camera on-site and then analyzed with the developed algorithms. A commercial system for greenhouses has also been developed, called the Scoutbox [17], which contains a high-resolution camera. Technicians carry the Scoutbox with them and manually place a yellow sticky card (YSC) in the box and take a picture. The cards are then analyzed by the company for whitefly counts.

CNN models can be classified on a higher level as a single-shot or two-shot object detection model. Deciding between the two is a crucial part of any object detection project. Whereas two-shot object detectors are very accurate, single-shot detectors are computationally faster while sacrificing accuracy. Two-shot detection models have two stages, the first being a region proposal, and a second stage which is a classification of those regions and refinement of the location prediction. Three popular two-shot detectors are Fast-RCNN, Faster-RCNN [18] and an extension of that called Mask-RCNN [19]. RCNN's were initially found to have drawbacks, primarily in slow object detection due to the need of feeding 2000 region proposals generated from selective search algorithm to the CNN every time, and, therefore, extensions of these models were developed. Fast-RCNN improves speed of operation by feeding the input image to the CNN to generate a convolutional feature map instead of feeding the regional proposals, which ensures the convolution operation to be done only once per image. Faster-RCNN eliminates the slow and time-consuming process of selective search and lets a separate network predict the region proposals, which make the model much faster than its predecessors. Faster-RCNN proposes regions in its first stage and uses the Fast-RCNN detector to identify objects in the proposed regions. Mask-RCNN is a branch added to Faster-RCNN that creates a binary mask to determine if a specific pixel in a bounding box is part of an object.

Single-shot detection skips the region proposal stage and yields final localization and content prediction at once. Elimination of the need for the region proposal network makes the single-shot detection process much faster than the two-stage processes. The more popular of these models are YOLO versions (You Only Look Once) [20] and RetinaNet [21]. As mentioned, accuracy is generally sacrificed for speed in these detection models. Single-shot detectors can also have problems with small and close images, both potential issues with whiteflies (approximately 1 mm in length), or other small insects.

Our University of Georgia research team has developed a system to detect, count and report whiteflies trapped on yellow sticky tape (YST) remotely and without maintenance for up to 4 months. The IoT web-based remote autonomous insect detector (Web-RAID) stations would be installed across cotton and cucurbit overlapping areas. A graphical user interface (GUI) would provide current and historical data on whitefly counts for all monitors in the field and include data graphing capabilities and data download in .csv format. This graphical user interface would help determine the migration pattern of whiteflies, providing farmers real-time data on when and where to apply management strategies.

One crucial step for the system to become operational is to develop the detection and counting algorithms that can detect and count whiteflies on digital images of YST taken inside a Web-RAID station using an 8 MP camera (Raspberry Pi V2 camera module, $1920 \times 1080$). The model selection depends on (1) how much data will need to be transmitted over cellular signal, (2) accuracy of the model and (3) size of the model and the cost associated with edge versus cloud computation. This information, together with the

temporal accumulation of whitefly counts and the location of stations, can provide the data necessary to understand and control whitefly populations.

Objectives:

1. Using a prototype Web-RAID station, collect whitefly images on yellow sticky tape in a rearing room with an NVIDIA Jetson compatible 8 MP camera for training and validating DNN models.
2. Use images to train two Deep CNN models, namely YOLOv4 and Faster-RCNN.
3. Evaluate the precision, accuracy and recall for detecting and counting whiteflies for both models.
4. Develop a web-based graphical user interface to display a graphical distribution of whiteflies measured by the DNN model.

## 2. Materials and Methods

### *2.1. Web-RAID (Web-Based Remote Autonomous Insect Detector) Station*

The RAID station is composed of cost-effective electronic and mechanical components. Costs for individual components, including utility box, computer, motor, camera, LED's and modem are approximately USD 400.00. Whiteflies are trapped on 5 cm wide × 12.5 cm exposed YST. The RAID station advanced the YST using a gear motor so that a fresh surface was available to capture whiteflies 2× a day at 11:00 AM and 6:00 PM. The exposed YST was pulled into the monitor, and six LED lights would illuminate the YST and an 8 MP Raspberry pi camera would take images of the tape. An on-board NVIDIA Jetson Nano computer would collect and send the image to an on-board data modem (Inseego, Model SYUS 160). The modem would send images to the webserver to run the trained models to count the number of whiteflies on the trap. Figure 1 shows a station and its components in the field.

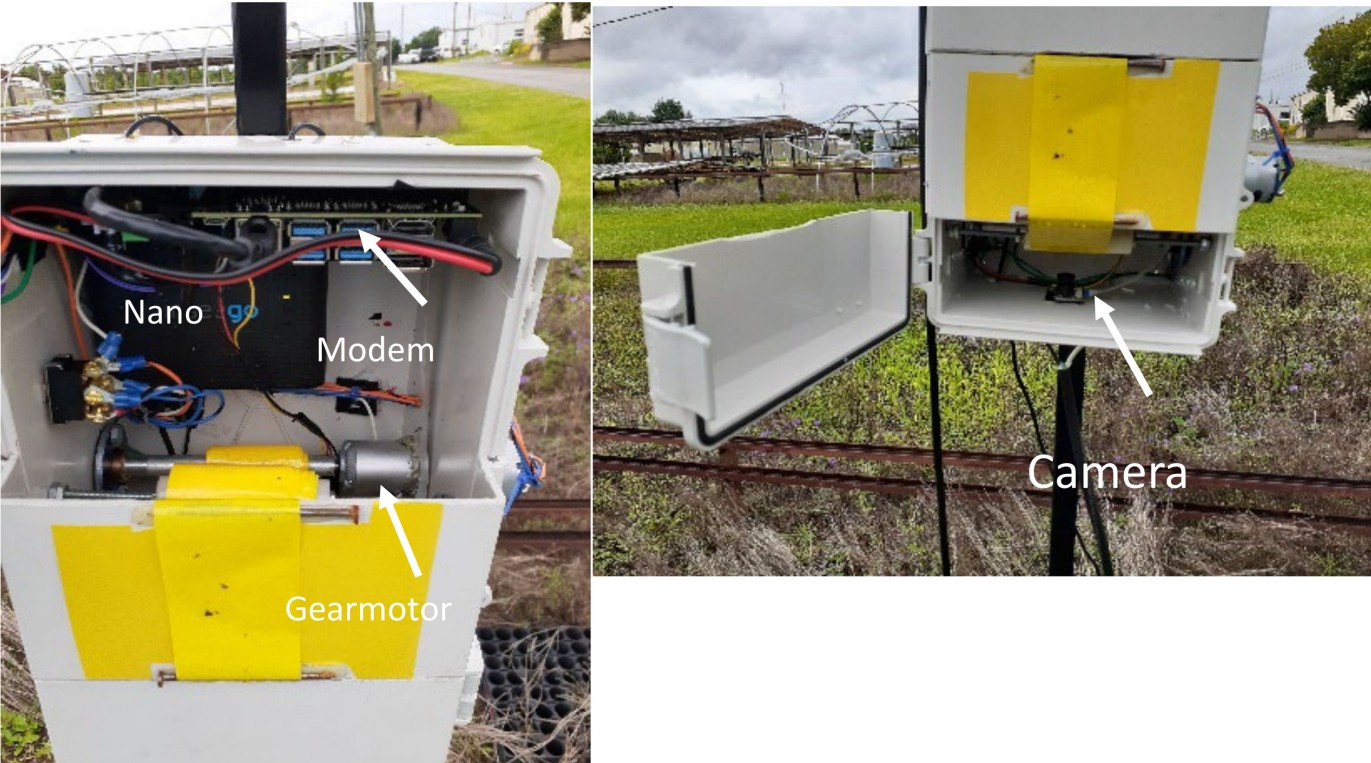

**Figure 1.** The web-RAID station opened at the top (**left**) and bottom (**right**) to see the main components.

We also required an IP address to provide secure communication between the web server and web clients (user). The communications protocol between client and server has been established using a client-server architecture. The client makes a connection request from the server and begins a client/server session that starts the read/write process of data transfer using TCP/IP protocol from the client in a remote location to the server at UGA laboratory office in Tifton, GA. The image is sent as packets of string characters and the server constructed back to the image after receiving the string and checksum.

### 2.2. The CNN Model

Figure 2 shows the workflow for selecting the model for the detection of whiteflies. The first step was to gather a dataset of whitefly images. Then, we chose two CNN models and labeled the dataset accordingly. In our case, we chose to compare a single-shot detector, YOLOv4 [22] for its speed, and Faster-RCNN [18], a two-shot detector, for its accuracy. The next step was to train both models and test those models to determine their accuracy, precision, F-1 score and recall. Based on the data comparing inference results, we choose the best model for our application.

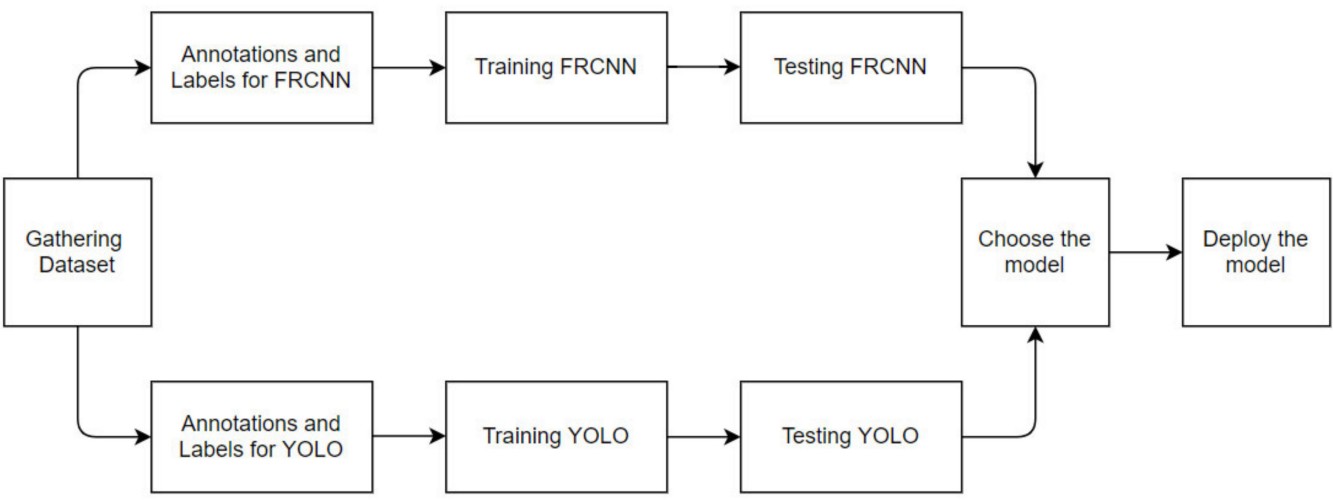

**Figure 2.** This figure shows the general workflow for model selection.

### 2.3. Data Collection

The whitefly images were taken using a whitefly monitoring station inside a controlled environment whitefly rearing room. Whiteflies were allowed to feed on cotton plant seedlings and provide multiple generations. Whiteflies are naturally attracted to the yellow color of the stations and images were collected two times per day at 11:00 AM and 6:00 PM. Each YST image contained around 60 to 80 whiteflies and resulted in over 8000 labels (Figure 3).

### 2.4. Model Selection

Object detection was performed with two different models, YOLOv4 and Faster-RCNN. YOLOv4 is a single-stage detector model, making it computationally small and fast. The second model we tested was Faster-RCNN. Faster-RCNN is a two-stage detector model that is resource-intensive, but is usually quite accurate. As whitefly detection is going to take place on static images, speed can be sacrificed for accuracy if needed.

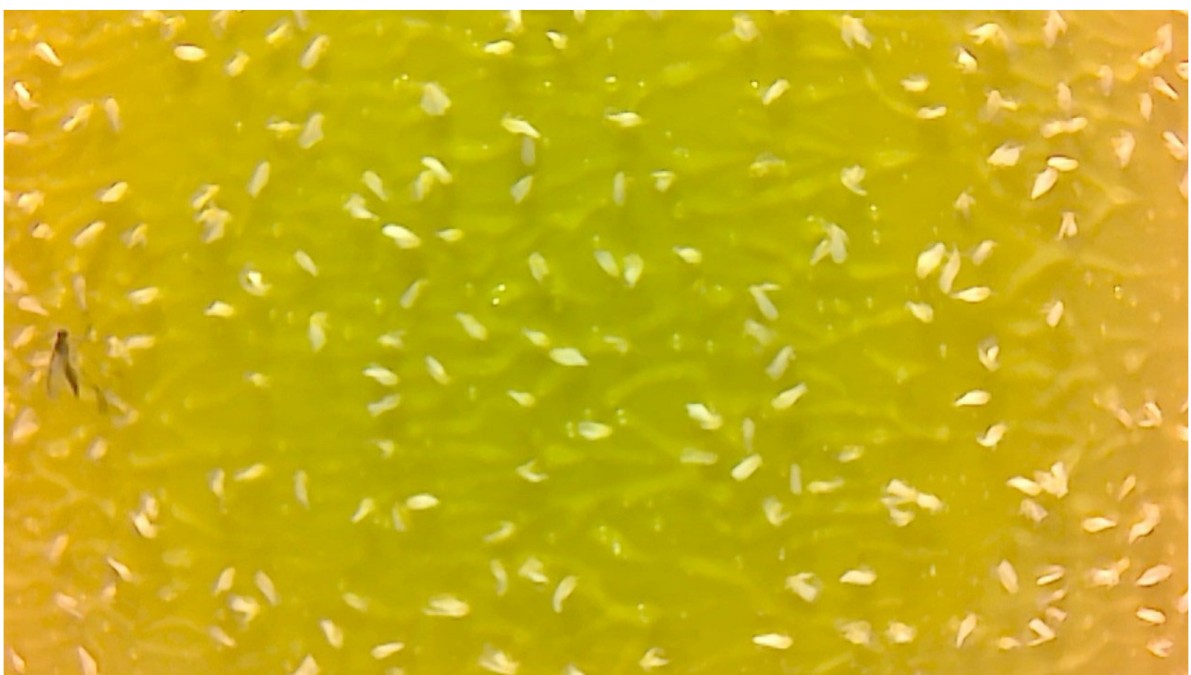

**Figure 3.** Whiteflies trapped on YST (yellow sticky tape).

*2.5. Data Processing and Labeling*

The images collected for training were 1920 × 1080 pixels (1080p). Each image was sub-sampled to a resolution of 640 × 640 pixels, which is a good size to save the image details and not hamper the time of training a model.

The central prerequisite for training on a pre-trained CNN model is labeling the images. We used LabelImg v1.8.6 (MIT License) to label all the images as it is very user-friendly. It permits the user to create bounding boxes around the target (whitefly body) of the image and afterward give a classification name as indicated by the user. Figure 4 shows the labeling done in the LabelImg tool. In our scenario, there are two classification labels, "Whitefly" and "others". The "Whitefly" label denotes the whiteflies in the image and the "others" label was used for negative training. Some white spots are formed because of the light reflecting from the sticky tape.

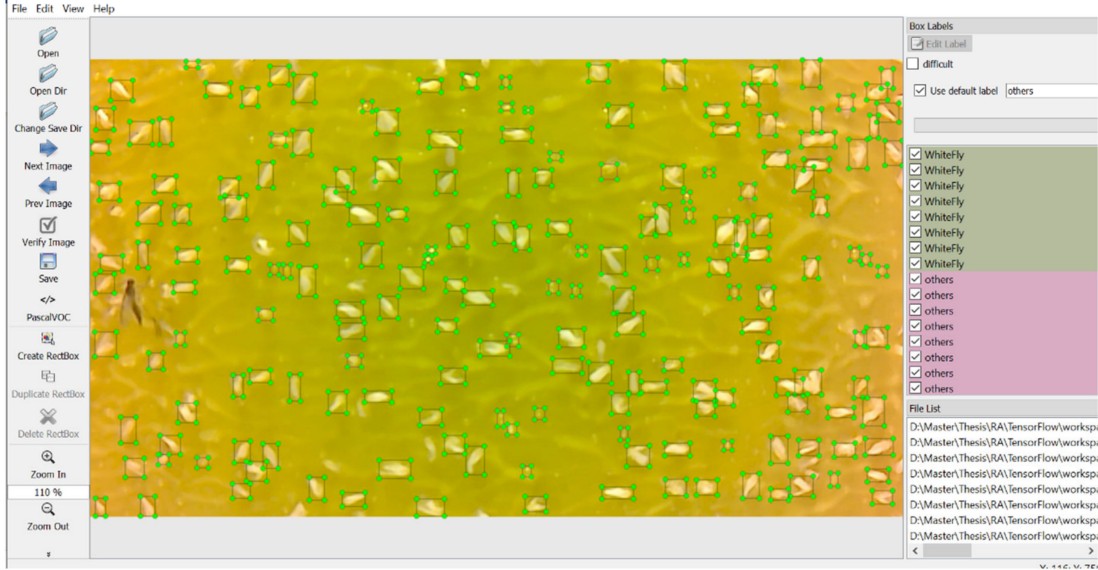

**Figure 4.** This figure shows the labeled image using LabelImg tool.

### 2.5.1. YOLOv4

Ninety percent of the dataset was used for training and the remaining 10% was used for testing the YOLOv4 model. Training data was 90% to increase the sample size for training. A file with ".names" extension is created to store the class name. Each line in the file is utilized per class name. For YOLOv4, the labels are stored in .txt format. Each label in YOLOv4 has five parameters. The first parameter is the line number in object names corresponding to the name of the object class (in our case, we have two classes, i.e., whitefly (0) and others (1)). The next two parameters are the object's center coordinates. The last two parameters are the width and height of the bounding box.

### 2.5.2. Faster-RCNN

Ninety percent of the dataset was used for training and the remaining 10% was used for testing the Faster-RCNN. Unlike YOLOv4, which uses .txt files to store the labels, Faster RCNN uses Pascal VOC .xml file to store its labels. Tensorflow uses the TFrecord file for training and testing. TFrecord file is a binary file format for the storage of data which has a significant impact on the performance of the import pipeline and therefore reduces the training time of the model. A python script was used to convert labels stored in a .xml file to train.tfrecord and test.tfrecord for training and testing, respectively. Additionally, as obj.names uses YOLOv4 for storing the classes, TensorFlow uses labelmap.pbtxt to store classes. Each line is utilized per class name.

### 2.6. Training and Testing

The training was conducted on a personal laptop with an AMD Ryzen 9 processor, 16 GB of RAM and a 6 GB RTX 2060 MAX-Q Graphics card. Initially, both the models were trained just to detect whiteflies. The testing showed that there were numerous false positives detected. Therefore, a new class label called "others" was created to train the model to recognize water droplets and light reflectance. This negative training (NT) did not significantly affect the YOLOv4 model performance, but Faster RCNN showed a drastic performance improvement.

Testing the model to the test data was accomplished on the same laptop as was used for training. The model was imported onto the webserver computer (8-Core Intel i7, 32 GB RAM, Nvidia GeForce GTX 1060 6 GB GPU, 1.5 TB HD) to analyze the images from the remote traps in the field.

### 2.6.1. YOLOv4

YOLOv4 is quite possibly the most visible and most popular real-time object detector available today. YOLOv4 is built using the Darknet framework. Darknet is an open-source neural network framework written in C and CUDA. For training our model, we used the pre-trained YOLOv4 weights (w). The maximum batch size was set to 6000. In total, 18 filters were used in the convolution layers. The steps were set to (4800, 5400). The steps determine the number of iterations at which scales will be applied. Scales are coefficients at which the learning rate is multiplied at a particular step. The training was run on the dataset for 4000 epochs, the loss started from 18.98 and reach 0.27. The training was stopped when the loss graph line was almost flat. Figure 5 shows the loss graph for YOLOv4. The Complete Intersection over Union (CIoU) loss [23] which is used to optimize loss regression is:

$$LOSS_{CIoU} = 1 - IoU + \frac{\rho^2\left(b, b^{gt}\right)}{c^2} + \alpha v \tag{1}$$

where $b$ and $b^{gt}$ are the central points of the bounding box, and bounding box ground truth ($B$, $B^{gt}$), $\rho^2\left(b, b^{gt}\right)$ is the Euclidean distance, $c$ is the diagonal length of the smallest enclosing box covering $B$ and $B^{gt}$ and $\alpha$, $v$ represents

$$\alpha = \frac{v}{1 - IoU + v} \tag{2}$$

$$v = \frac{4}{\pi^2}\left(\arctan\frac{w^{gt}}{h^{gt}} - \arctan\frac{w}{h}\right)^2 \tag{3}$$

$w$ and $h$ are the width and height of the bounding box, respectively. The Total Loss function for YOLOv4 is given by:

$$
\begin{aligned}
Total_{LOSS} = {} & 1 - IoU + \frac{\rho^2(b, b^{gt})}{c^2} + \alpha v \\
& - \sum_{i=0}^{S^2}\sum_{j=0}^{B} I_{ij}^{obj}\left[\hat{C}_i\log(C_i) + (1 - \hat{C}_i)\log(1 - C_i)\right] \\
& - \lambda_{noobj}\sum_{i=0}^{S^2}\sum_{j=0}^{B} I_{ij}^{noobj}\left[\hat{C}_i\log(C_i) + (1 - \hat{C}_i)\log(1 - C_i)\right] \\
& - \sum_{i=0}^{S^2} I_{ij}^{obj}\sum_{c\in classes}\left[\hat{p}_i(C)\log(p_i(C)) + (1 - \hat{p}_i(C)\log(1 - p_i(C))\right]
\end{aligned} \tag{4}
$$

$C$ is Class, $S$ is the number of grid cells and $p$ is the probability that Class is detected.

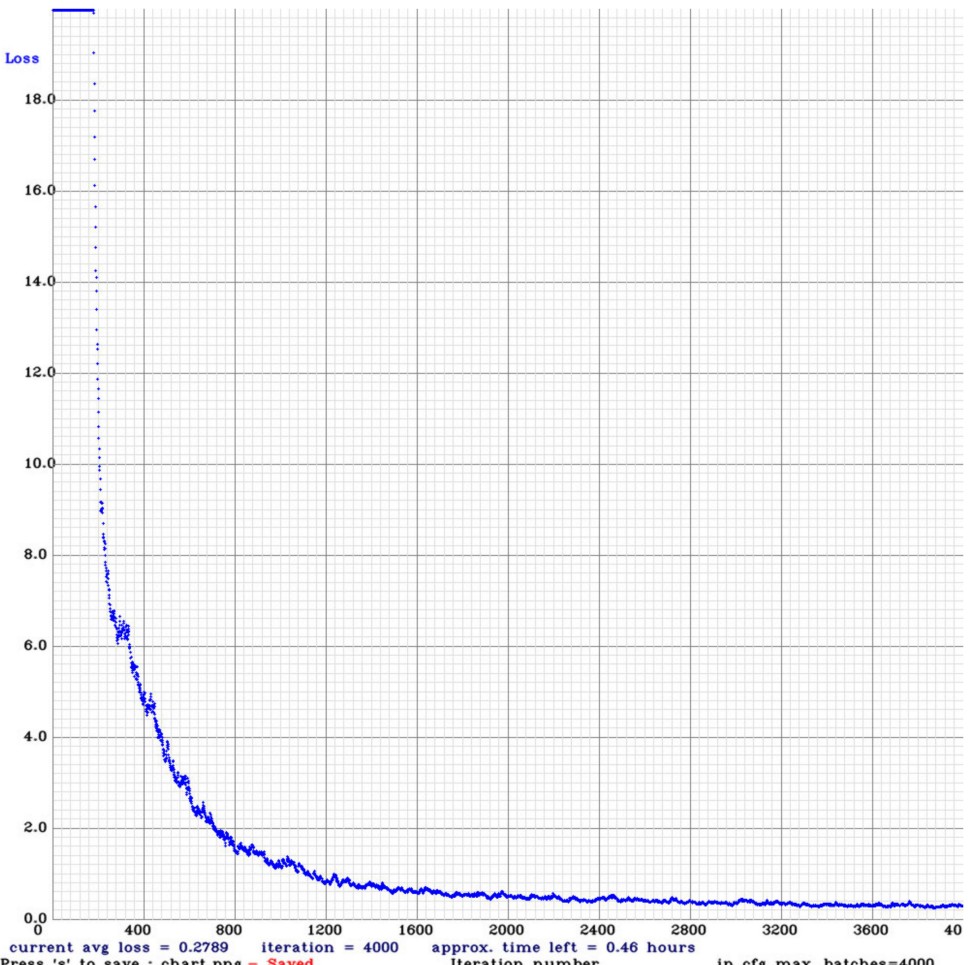

**Figure 5.** YOLOv4 Loss Graph.

### 2.6.2. Faster-RCNN

Google's Object Detection API was used to train the pretrained Faster-RCNN model from Google's Model Zoo. This model was originally trained on the coco dataset. The loss function of Faster RCNN for any image is defined as

$$L\left(\{p_i\}, \{t_i\}\right) = \frac{1}{N_{cls}} \sum_i L_{cls}(p_i, p_i^*) + \lambda \frac{1}{N_{reg}} \sum_i p_i^* L_{reg}(t_i, t_i^*) \tag{5}$$

Here, $i$ is the index of an anchor in a mini-batch and $p_i$ is the predicted probability of anchor $i$ being an object. The ground-truth label $p^*_i$ is **1** if the anchor is positive and 0 if the anchor is negative. $t_i$ is a vector representing the 4 parameterized coordinates of the predicted bounding box, and $t^*_i$ is that of the ground-truth box associated with a positive anchor. The classification loss $L_{cls}$ is log loss over two classes (object vs. not object). $L_{reg}$ is the regression loss. Figure 6 shows the Loss Graph for Faster-RCNN.

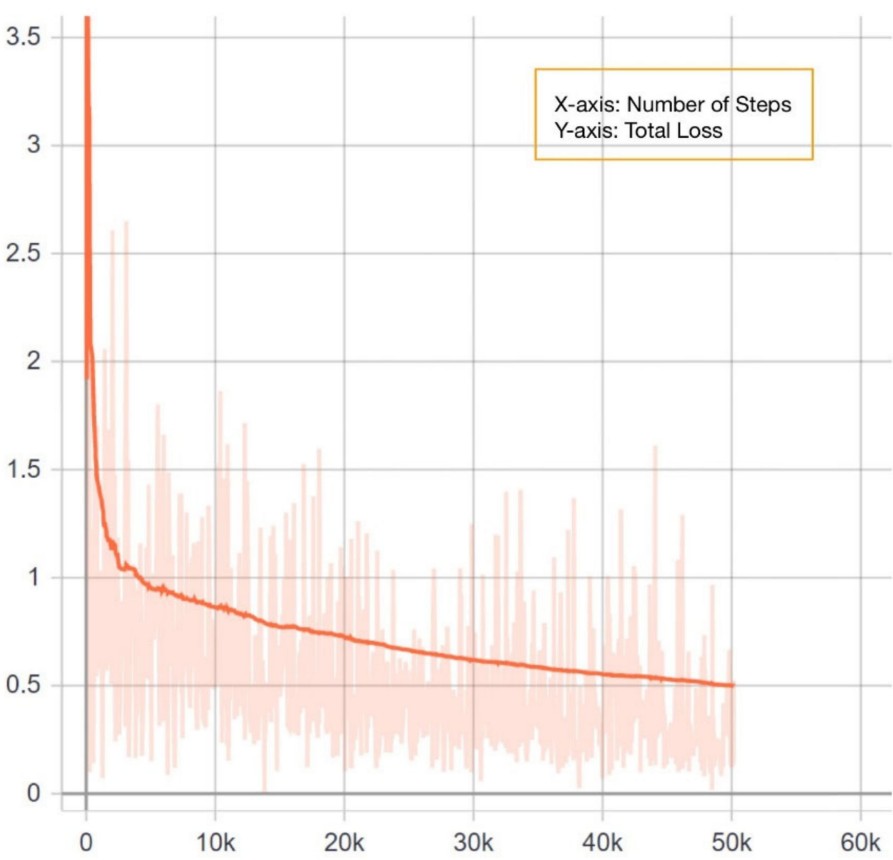

**Figure 6.** Faster-RCNN Loss Graph.

### 2.7. The Web-RAID Monitoring System

The Web Application, called "Web-Raid Monitor System", was designed to help farmers and researchers to visualize the data regarding whiteflies population and also help analyze their migration pattern. A Jetson Nano computer processor with its camera, the Faster-RCNN model running on the server and the web application are all integrated to give users a seamless and completely automated experience. Figure 7 shows the flow of this System.

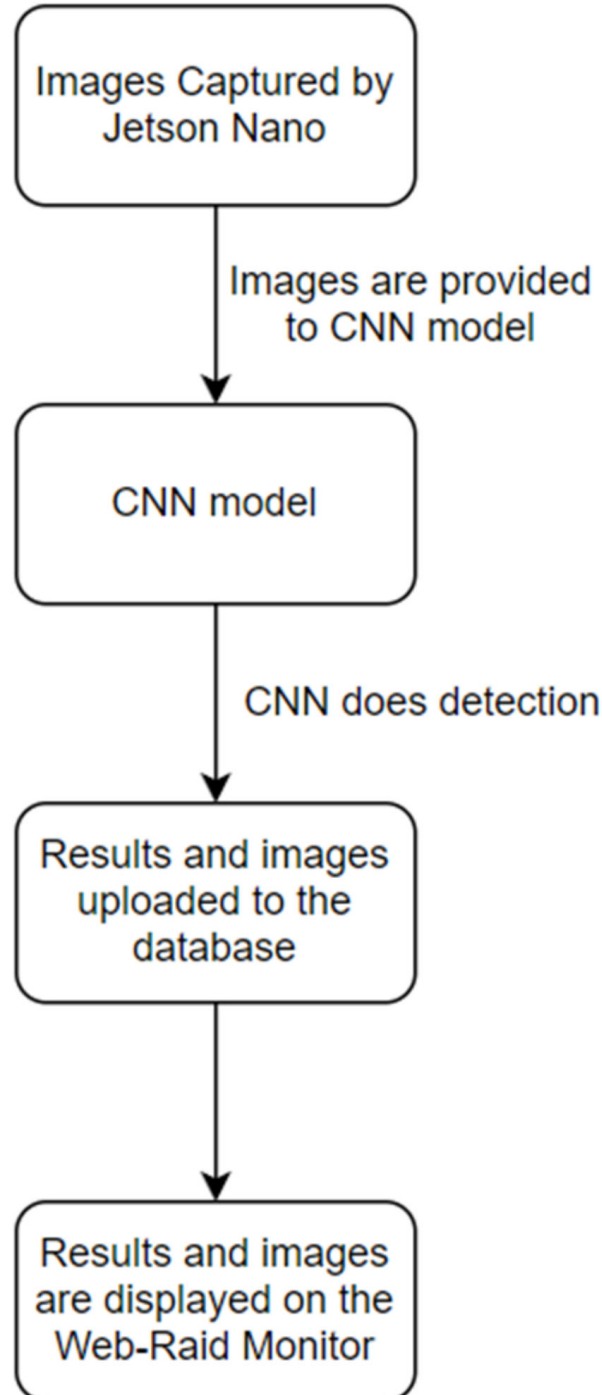

**Figure 7.** General Flow of Web-Raid Monitor System.

Each Web-RAID station consists of a Jetson Nano computer, 8-mb Nano compatible camera, gearmotor, LED lights and a modem. The Nano manages the gearmotor to advance the yellow sticky tape so that tape that has attracted whiteflies is pulled into the station until in front of the camera. Then, the LED lights are turned on and the camera takes a digital image of the sticky tape. The image is sent to the modem (Inseego SKYUS 160), which in turn sends it to a static IP address hosting the web server. Then, the Faster-RCNN model detects and counts the number of whiteflies in these images. The data is then stored in the database connected to the Web-Raid Monitor System. This web app was developed using the Django Web framework. Programming languages Python, HTML5 and Javacript were used to build the app. The web application consisted of the following components:

1.     Web-Raid Monitor

Figure 8 shows the home page of our web application with a bubble map of Georgia that was implemented using JavaScript (HighCharts). The bubbles indicate station locations in the state. When the cursor is placed over the bubble, it provides the location and count of whiteflies detected to date. As the count increases or decreases, the size of the bubble also changes accordingly. This helps provide a quick high-level visual of migration pattern of the whiteflies to the user. It also provides links to the original image, analyzed image with labeled whiteflies and a count of the whiteflies between a particular time frame of a particular location. Figure 9 shows the image link and the count of whiteflies in that image sorted according to user-specified date and location.

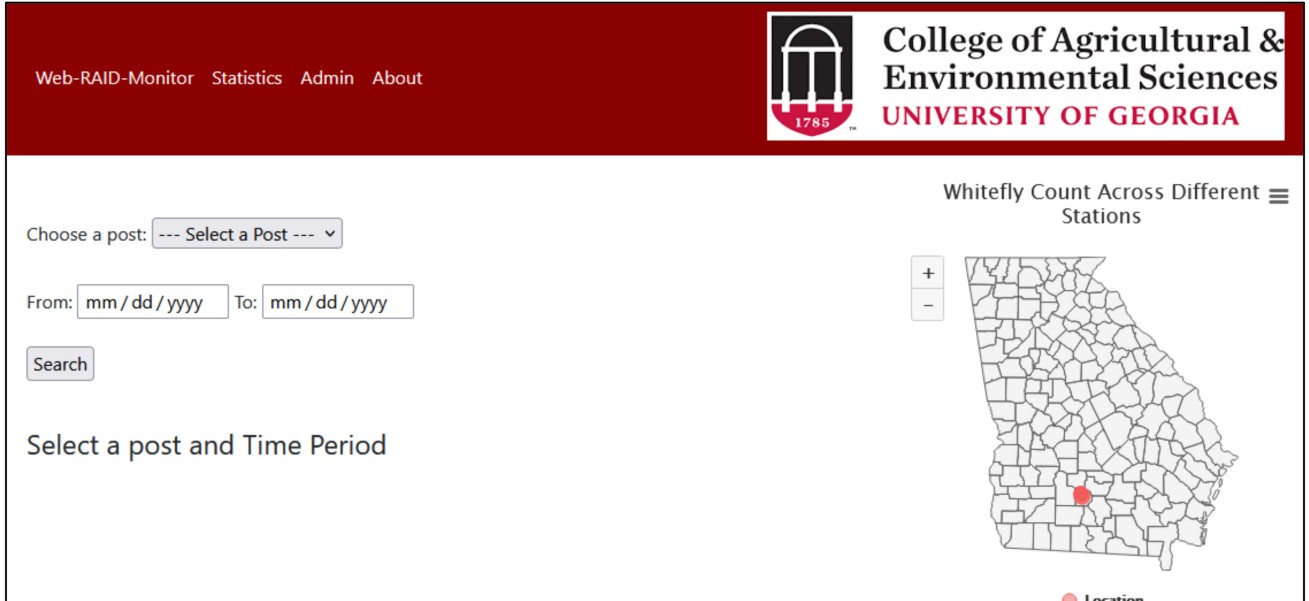

**Figure 8.** Web-Raid Monitor System homepage.

2.     Statistics

Figure 10 shows a graphical view of the number of whiteflies counted across all locations. The green bar shows the total count of all images taken in the morning and the red bar is the reading taken in the evening. Data can be filtered by the results according to the dates. Results can also be downloaded in .cvs format.

3.     Admin Panel

The website administration panel is where the data that can be displayed on the webpage is managed without manually firing any query to update data in the database. The admin panel provides an interface to directly add/delete different location/posts or manually add pictures of whiteflies and counts for a particular date (in case there is some problem for the web app to communicate with the server).

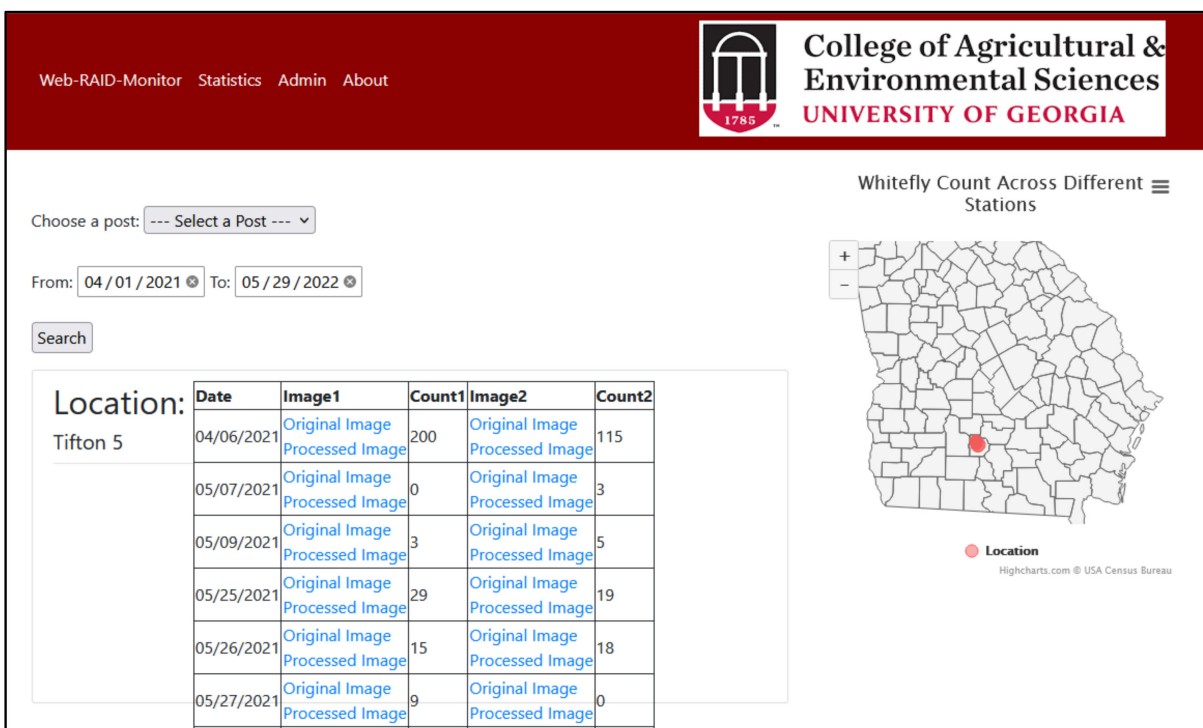

**Figure 9.** Results sorted according to date and location.

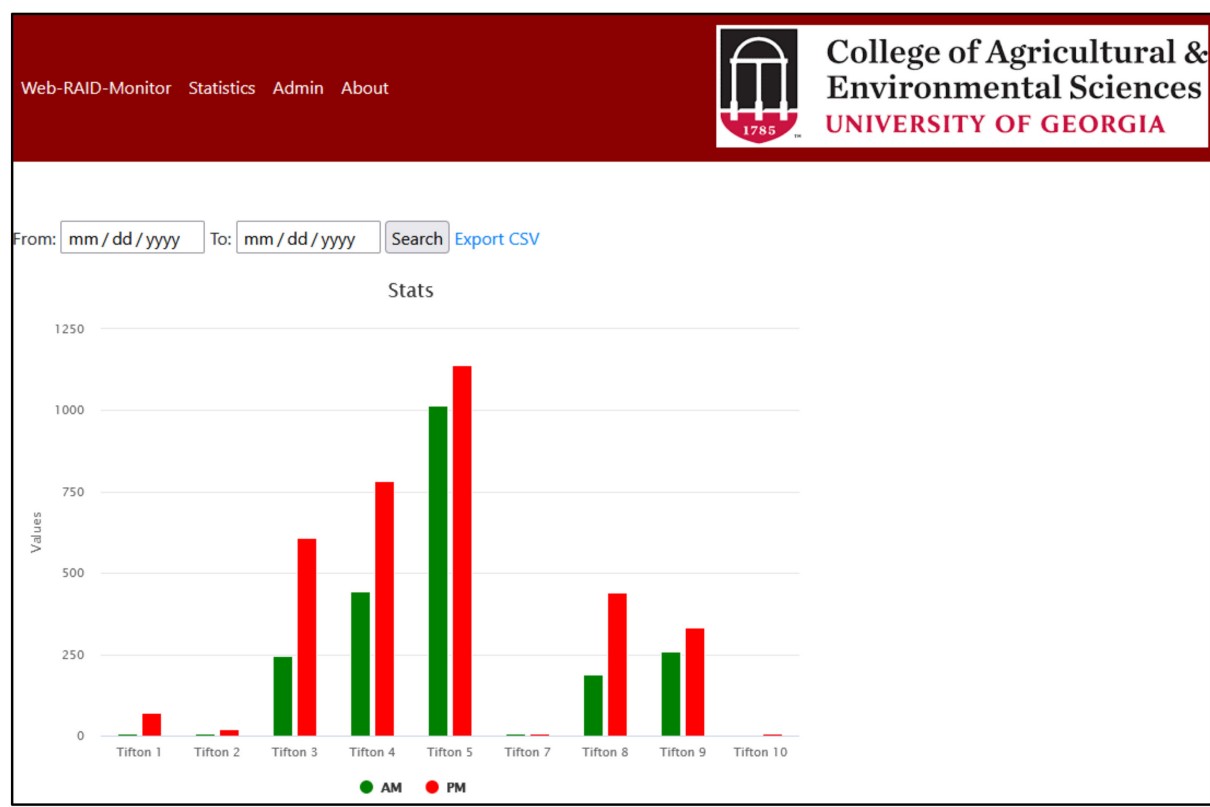

**Figure 10.** Bar chart of whitefly counts at each station for AM and PM.

### 3. Result

*3.1. Negative Training*

The models were trained just on the whitefly labels. Unlike YOLOv4, the trained Faster-RCNN model had a significant number of false positives. This occurred due to the

imperfect placement of the LED's illuminating the yellow-sticky tape when the image was taken by the camera as well as the slight angle of the tape to the camera. Figure 11a shows the false positive detections by the Faster RCNN model. To overcome this issue, negative training was performed to differentiate the white spots from whiteflies. Figure 11b shows the model result after negative training.

The YOLOv4 model had high precision right at the beginning because it did very little false positive detection. Even after negative training, the results were almost the same (Table 1).

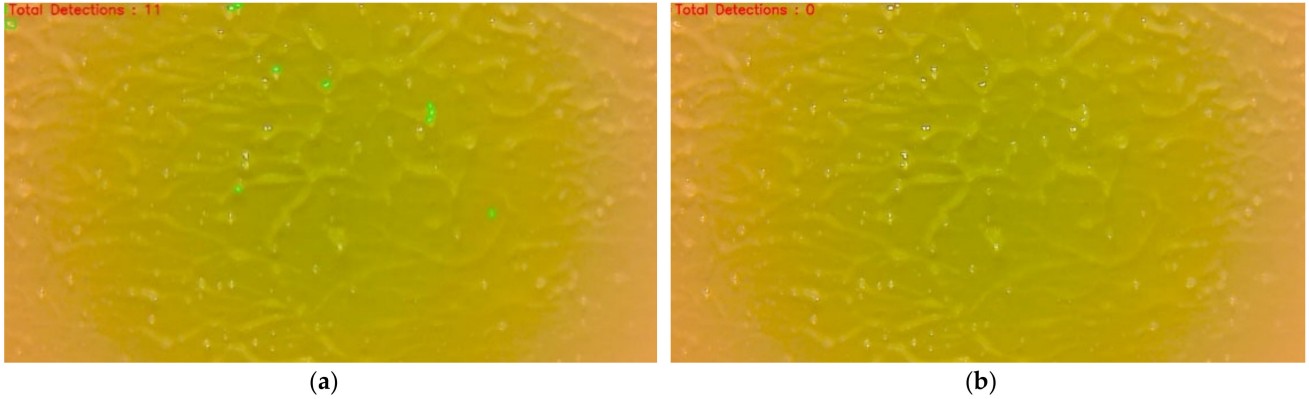

(**a**)                                                                  (**b**)

**Figure 11.** (**a**) Results of Faster RCNN before negative training and (**b**) Results of Faster RCNN after negative training.

**Table 1.** Performance metrices.

| | Mean Average Precision (IoU = 0.5) | Accuracy | F-1 Score | Recall | Mean Inference Time (s) |
|---|---|---|---|---|---|
| Faster-RCNN (NT) | 96.29% | 95.08% | 0.96 | 98.69% | 23.72 |
| Faster-RCNN | 65.19% | 60.48% | 0.75 | 89.32% | 23.56 |
| YOLOv4 (NT) | 97.16% | 71.77% | 0.83 | 73.31% | 8.94 |
| YOLOv4 | 96.70% | 71.41% | 0.82 | 73.19% | 8.78 |

*3.2. Experimental Analysis*

The sensitivity or recall, accuracy, F1 score and precision of the models were determined as follows:

$$\text{Recall} = \text{TP}/\text{TP} + \text{FN} \qquad (6)$$

$$\text{Accuracy} = \text{TP} + \text{TN}/\text{TP} + \text{FP} + \text{FN} + \text{TN} \qquad (7)$$

$$\text{Precision} = \text{TP}/\text{TP} + \text{FP} \qquad (8)$$

$$\text{F1 Score} = 2 * (\text{Recall} * \text{Precision})/(\text{Recall} + \text{Precision}) \qquad (9)$$

where TP is the number of true positives detected by the model, FN is the number of false negatives detected by the model, TN is the number of true negatives and FP is the number of false-positive labels in the given dataset.

Testing revealed that the two-shot detector, Faster-RCNN, was more accurate and more suitable for our application. Faster-RCNN obtained an accuracy of 95.08%, whereas YOLOv4 obtained an accuracy of 71.77%. Table 2 shows all the performance matrices for Faster-RCNN with Negative Training (NT), Faster-RCNN without negative training, YOLOv4 with Negative training (NT) and YOLOv4 without Negative Training. Faster-RCNN (NT) had the best F-1 Score, Recall and accuracy.

**Table 2.** Faster-RCNN Confusion Matrix.

| | **True Positive** | **False Positive** | **False Negative** | **True Negative** |
|---|---|---|---|---|
| Faster-RCNN (NT) | 832 | 32 | 11 | 0 |
| FasterRCNN | 753 | 402 | 90 | 0 |

Figures 12 and 13 show the detection results of Faster-RCNN and YOLOv4 respectively deployed on the Web-Raid-Monitor system for two images. Tables 2 and 3 show the confusion matrices for Faster-RCNN and YOLOv4, respectively.

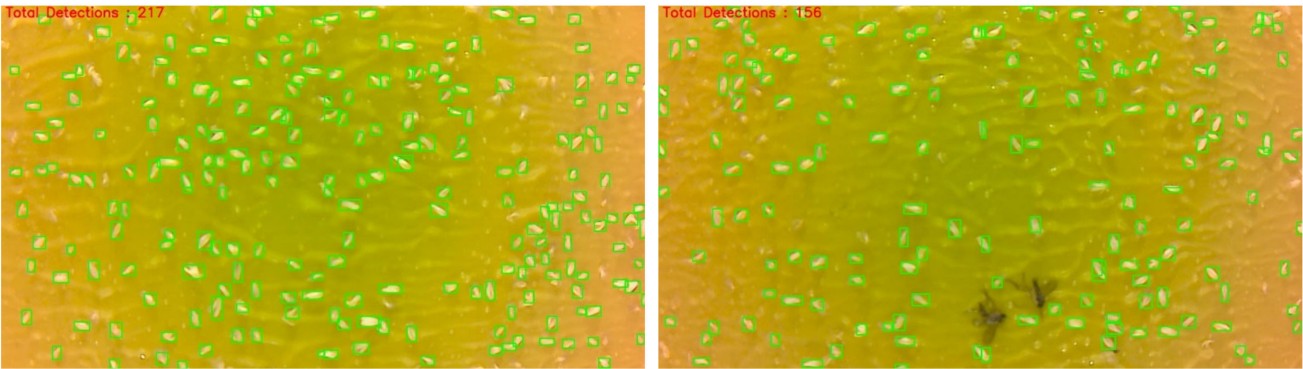

**Figure 12.** Two examples of detection results forFaster-RCNN model with negative training using yellow sticky tape image from web-RAID monitor.

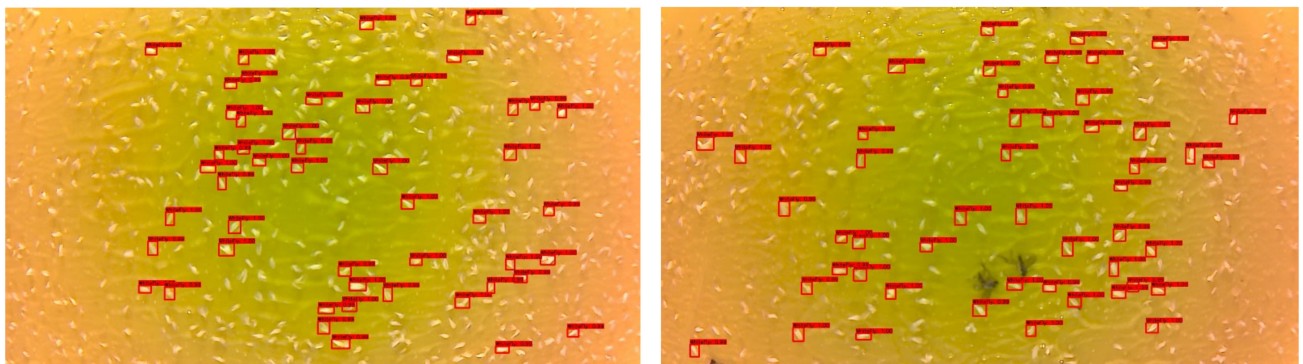

**Figure 13.** Two examples of detection results for YOLOv4 model with negative training using yellow sticky tape image from web-RAID monitor.

**Table 3.** YOLOv4 Confusion Matrix.

| | **True Positive** | **False Positive** | **False Negative** | **True Negative** |
|---|---|---|---|---|
| YOLOv4 (NT) | 618 | 18 | 225 | 0 |
| YOLOv4 | 617 | 21 | 226 | 0 |

## 4. Conclusions

The developed Web-Raid-Monitor system will be very useful in determining the migration pattern of whitefly insects. The Faster-RCNN model deployed on the Web-Raid-Monitor system accurately detects and counts whiteflies. The visual representation (Bubble Map, Graphs) provided on the Web-Raid web app will help farmers and researchers to take precautionary measures against whitefly infestation. From the experimental analysis, the accuracy of Faster-RCNN was 95.08% and that of YOLOv4 was 71.77%. YOLOv4 being a single-shot detector is small and compact with inference time of 8.94 s to that of 23.72 s for

the Faster RCNN, but Faster-RCNN is more accurate and better suited for this application since the image analysis is not required to be real-time and can be conducted in the cloud. Adding new species of insects, in the future, will be simply done by using additional training datasets of insects on sticky tape. The Web-Raid-monitor stations can then detect and count multiple species attracted to the YST.

**Author Contributions:** Conceptualization, G.C.R., D.R. and J.M.S.; methodology, G.C.R., D.R., C.U.P., C.M., M.H. and K.F.; software, G.C.R., C.U.P., K.F. and C.M.; validation, G.C.R., C.U.P., M.H. and S.B.; formal analysis, C.U.P. and G.C.R.; investigation, C.U.P., C.M., M.H., J.M.S., D.R., K.F. and S.B.; resources, G.C.R., D.R. and J.M.S.; data curation, G.C.R.; writing—original draft preparation, C.U.P. and G.C.R.; writing—review and editing, C.U.P., C.M., M.H., J.M.S., D.R., K.F. and S.B.; visualization, G.C.R.; supervision, G.C.R., D.R. and J.M.S.; project administration, G.C.R., D.R. and J.M.S.; funding acquisition, G.C.R., D.R. and J.M.S. All authors have read and agreed to the published version of the manuscript.

**Funding:** This work was funded in part by the USDA Non-Assistance Cooperative Agreement #58-6080-9-006 "Managing Whiteflies and Whitefly-transmitted Viruses in Vegetable Crops in the Southeastern U.S.".

**Informed Consent Statement:** Not applicable.

**Data Availability Statement:** Data will be made available on a newly developed whitefly team webpage, and later on the Web-Raid-Monitor system.

**Acknowledgments:** We thank Melissa Thompson for helping with sticky cards, Jermaine Parker for helping with sticky cards and colony of whiteflies produced. Studies that contributed to our understanding of this subject matter were conducted using the facilities of the Georgia Agricultural Experiment Stations.

**Conflicts of Interest:** The authors declare no conflict of interest.

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
