# Peer review of "Comparison of Single-Shot and Two-Shot Deep Neural Network Models for Whitefly Detection in IoT Web Application"

_agriengineering, doi:10.3390/agriengineering4020034_

Round 1

Reviewer 1 Report

The proposed method has been argued to provide innovation in the field of agriculture, but similar systems have not been adequately examined in terms of hardware, software, or cost. To emphasize the scientific contribution, the literature review should be done more rigorously.

The subject mentioned in the article is a very simple application for those who work in image processing. Would a very simple blob detection or a simple filter design be sufficient for the study? Why weren't they used? It should be explained and emphasized why the basic methods are not preferred.

The YOLO and FastRCNN presented are architectures that are already out of the box, so the authors are just the user, not the DNN designer. In this case, they are expected to add at least 2 or 3 more systems from alternative architectures. In this way, the performances of different techniques will be compared, and the scientific contribution will be increased. In comparison, metrics such as train time, accuracy, F1 metrics and response time / frame rate must be given. In addition, presenting the confusion matrix is important in terms of verifying the metrics.

Reviewer 2 Report

The author are describe an interesting approach on an early-warning system for insects, while two different neural network approaches are evaluated. The work is interesting but has several flaws:

The main points of this research should be communicated in a more accurate and specific manner.

References to similar and/or preexisting work are quite few.    

The objectives pointed out in the Introduction section are more general than the title of the article.

Several typo and syntax errors should be corrected and the authors in their descriptions should avoid to use 1st or 2nd person, and have instead a more neutral and formal style.

More details should be given on the specific software used for train each CNN model, especially for the YOLO case. 

The inclusion/exclusion of other insects from the “others” class should be better justified.   

The characteristics of the server computer being used for the continuous classification of the whiteflies are (possibly) the same with the one being used for training the models, but this is not clearly stated into the document.

Some Sections very short while Section 3.7 is more like a user manual than part of a research article.

The communication part of the whole system (i.e., among the field and the central modules) is poorly explained.

Financial cost information for both stations and central parts should be very welcome.      

The authors should also explain why not use more samples or an 80/20 ratio between training and testing samples.

A more mature version of this research should be submitted.

Round 2

Reviewer 1 Report

Confusion matrix representations were in bad shape in the pdf file I received. Once corrected, it's okay to accept it.

Reviewer 2 Report

The authors, in their revised manuscript, incorporated the reviewer’s suggestions in a quite satisfactory degree, thus only minor issues should be addressed:

1) Text font size and formula description should follow the MDPI manuscript formatting specifications.

2) More referencing material should be welcome.

3) Short descriptions should be further enriched with details for the non-expert reader.   
